# Nodule Inception Is Not Required for Arbuscular Mycorrhizal Colonization of *Medicago truncatula*

**DOI:** 10.3390/plants9010071

**Published:** 2020-01-06

**Authors:** Anil Kumar, Donna R. Cousins, Cheng-Wu Liu, Ping Xu, Jeremy D. Murray

**Affiliations:** 1National Key Laboratory of Plant Molecular Genetics, CAS-JIC Centre of Excellence for Plant and Microbial Science (CEPAMS), CAS Center for Excellence in Molecular and Plant Sciences, Institute of Plant Physiology and Ecology, Chinese Academy of Sciences, Shanghai 200032, China; anilkumar.ptu@gmail.com; 2Cell and Developmental Biology, John Innes Centre, Norwich Research Park, Norwich NR4 7UH, UK; donnacousins50@gmail.com (D.R.C.); chengwu.liu@slcu.cam.ac.uk (C.-W.L.); 3Shanghai Engineering Research Center of Plant Germplasm Resource, College of Life Sciences, Shanghai Normal University, Shanghai 200234, China

**Keywords:** common symbiosis genes, *Sinorhizobium meliloti*, *Rhizophagus irregularis*

## Abstract

Most legumes can engage in symbiosis with N-fixing bacteria called rhizobia. This symbiosis, called nodulation, evolved from the more widespread symbiosis that most land plants form with arbuscular mycorrhiza, which is reflected in a common requirement of certain genes for both these symbioses. One key nodulation gene, *Nodule Inception* (*NIN*), has been intensively studied. Mutants in *NIN* are unable to form nodules, which has made it difficult to identify downstream genes under the control of NIN. The analysis of data from our recent transcriptomics study revealed that some genes with an altered expression of *nin* during nodulation are upregulated in mycorrhizal roots. In addition, another study reported the decreased colonization of *nin* roots by arbuscular mycorrhiza. We therefore investigated a role for NIN in mycorrhiza formation. Our time course study, using two *nin* alleles with differing genetic backgrounds, suggests that that loss of NIN does not affect colonization of *Medicago truncatula* roots, either in the presence or absence of rhizobia. This, and recent phylogenetic analyses showing that the loss of NIN is correlated with loss of nodulation in the FaFaCuRo clade, but not with the ability to form mycorrhiza, argue against NIN being required for arbuscular mycorrhization in legumes.

## 1. Introduction

Arbuscular mycorrhization is a beneficial symbiosis formed between arbuscular mycorrhizal (AM) fungi and many terrestrial plants. It is believed to be older than 400 million years old, and its retention in over 70% of land plants suggests it provides a strong selective advantage to its hosts [1]. This symbiosis promotes the uptake of phosphates and other nutrients by the plant host at the expense of host carbon, supplied to the fungal endosymbiont. It entails the constant exchange of signals between the host and symbiont, which ultimately leads to the formation of nutrient exchange and fungus-accommodation structures called arbuscules within root cortical cells [2]. This process requires the differential activation of hundreds of genes which have been studied at the evolutionary and functional genetic levels using genomic approaches [3].

A second widespread symbiosis, called nodulation, can be formed by most species belonging to the third largest plant family, the legumes (Fabaceae) with gram-negative soil bacteria collectively called rhizobia [4]. The nodulation of most legume plants involves the intracellular infection of root hairs by rhizobia followed by the colonization of the cortical layers of the root [5]. This symbiosis leads to the formation of special root outgrowths called nodules. Nodules are comprised mainly of cells filled with nitrogen-fixing rhizobia contained within membrane-bound structures called symbiosomes [6]. Like mycorrhization, nodulation requires ongoing signal-exchange with the rhizobia. Plant-released flavonoids trigger the production of a counter-signal by the rhizobia, a mixture of lipo-chito oligosaccharide signal molecules called Nod factors [7,8]. Nod factors are perceived by host Nod factor receptors which trigger the increased expression of several transcription factors, including Nodule Inception (NIN), ERF Required for Nodulation1 (ERN1), NFYA1, and the GRAS transcription factors Nodulation Signalling Pathway1 (NSP1) and NSP2 [9,10,11,12,13,14,15,16,17,18]. The activation of these, and other transcription factors, results in extensive transcriptional changes that lead to the formation of nitrogen fixing nodules.

Since its discovery, Nodule Inception (NIN) has been extensively studied for its role in nodulation [9]. Mutants lacking NIN respond to rhizobia by root hair deformation, but do not initiate the formation of infection threads, nor do they form nodules [9,10,19]. NIN is the founding member of a small family of transcription factors, called NIN-like proteins (NLPs), that are present in all plants and have homologs in algae [20]. NLPs in *Arabidopsis* have demonstrated roles in nitrate sensing, uptake and assimilation [21]. In legumes, two NLPs have been implicated in the nitrate suppression of nodulation [22,23], suggesting that NIN’s function in positively controlling infection by rhizobia is unique within the NLP family. Interestingly, almost all members of the Fabales, Fagales, Cucurbitales and Rosales (FaFaCuRo) clade that have lost the ability to nodulate have also lost *NIN*, along with many other nodulation-specific genes [24,25].

Despite NIN’s importance, the identification of the genes downstream of NIN was impeded by the complete lack of nodules on *nin* mutants. However, in recent years, direct targets of NIN were identified using chromatin immunoprecipitation in *Lotus japonicus* and the NIN regulon was characterized using a root-hair transcript profiling approach [26]. Amongst NIN’s targets are *Nodulation Pectate Lyase 1* (*NPL1*) [27] which is required for rhizobial infection and at least two CCAAT-box transcription factors that are associated with nodule organogenesis [16,17,18,28]. Indeed, NIN’s regulon was estimated to include at least 120 genes [26]. Our analysis of *nin*’s root hair infectome found many genes with deregulated expression in *nin*, relative to wild type controls, that are also upregulated in roots colonized by AM fungi. Furthermore, a recent paper found decreased mycorrhizal colonization in *M. truncataula nin-1* [29]. Based on this, we investigated a potential role for NIN in nodulated and non-nodulated roots of *M. truncatula*.

## 2. Results

### 2.1. nin Mutants Have No Obvious Mycorrhizal Phenotype

A comparison of genes that were differentially expressed in root hairs of rhizobia-inoculated seedlings (*nin-1* vs wild type) to those genes induced by *Rhizophagus irregularis* revealed some overlap (Appendix A). About 40 genes were found with decreased expression in *nin* and increased expression during AM colonization, and about 50 others were upregulated in both *nin* and mycorrhizal roots. To determine whether this could influence colonization by arbuscular mycorrhiza, either with or without nodulation, an experiment was set up using *nin-1* and *nin-2,* that are from the *M. truncatula* cv Jemalong A17 and *M. truncatula* ssp. *tricycla* R108 backgrounds, respectively. Both *nin* alleles were scored over a time course of 2, 3, 4 and 5 weeks post inoculation (wpi) with *R. irregularis* DAOM197198 (10% chive inoculum). Half the plants were inoculated with *Sinorhizobium meliloti* Rm1021. Ten plants were grown for each genotype/time point/experimental condition and root samples were harvested. Nodules were completely absent from both *nin* mutants in all conditions, and non-inoculated wild type plants. Rhizobia-inoculated wild type plants nodulated normally. The root samples were then stained and scored for arbuscules (Figure 1). Although some reductions in AM colonization were observed in certain treatments (Figure 1b), overall the results showed no consistent difference in AM colonization between WT and either *nin* mutant, with or without nodulation.

### 2.2. NIN Expression during Mycorrhization

During nodulation, the *NIN* transcript levels are strongly increased, especially in the root hairs of rhizobia infected plants and nodules [30,31]. To determine if *NIN* is transcriptionally induced during mycorrhization, we compiled data from several RNA-seq and microarray transcriptome studies. The data were downloaded from the *M. truncatula* and the *Lotus japonicus* Gene Expression Atlas [32,33]. No difference was found in *NIN* expression in arbuscule cells vs adjacent cells or uncolonized cortical cells of *M. truncatula*, collected using laser capture microdissection (LCM) 3 weeks after inoculation with *R. irregularis* [34] (Figure 2a). A second dataset indicated that *NIN* expression was unchanged in 6-week-old uninoculated vs *R. irregularis* inoculated *M. truncatula* roots [35] (Figure 2a). To broaden our analysis, *NIN* expression was examined using data from a similar experiment carried out with another model legume, *L. japonicus* [36]. Similar to our findings in *M. truncatula*, no significant difference in *NIN* expression was detected between uninoculated and *Gigaspora margarita* inoculated roots at either of the two time points studied (Figure 2a). Finally, we used RNAseq data normalized across two different experiments to compare *NIN* expression in nodulated and mycorrhized roots, using the mycorrhiza-induced transcription factor *RAM1* as a reference [37,38]. As expected, an increase in *NIN* expression of two orders in magnitude was observed in nodules relative to control roots (Figure 2b). In mycorrhizal roots, *RAM1* was highly induced, but *NIN* expression was similar to baseline levels, and as before no induction of *NIN* in mycorrhizal roots relative to mock-inoculated controls was observed.

## 3. Discussion

We investigated *nin* for a potential mycorrhizal colonization phenotype, but no consistent effect, either increased or decreased, was found. This suggests that *NIN* doesn’t play a direct or indirect role in mycorrhization. The enhancement of certain mycorrhiza-induced genes in the root hairs of rhizobia-inoculated *nin* seedlings may be a consequence of a loss of negative feedback, leading to the increased activation of *NIN*-independent infection gene expression, which includes several common symbiotic genes induced both by mycorrhizal and rhizobia. *NIN* controls at least two negative regulators of infection in root hairs, *CLE12/CLE-RS2* [40,41] directly, and several gibberellic acid biosynthesis genes [26], which act together to limit infection.

While our conclusion regarding the absence of an AM phenotype is consistent with earlier reports on *nin* in *L. japonicus* [19,42,43], it contradicts an earlier study that found that *M. truncataula nin-1* had strongly reduced AM colonization at nine weeks post inoculation, and had a mild decrease in penetration events at two weeks post inoculation [29]. The differences could be accounted for by the use of different types of inoculum used. The previous study used a fixed quantity of spores as inoculum, while the inoculum used for this study was a fresh inoculum containing spores, mycelia and colonized root fragments which presumably more closely resembles what plants encounter in their natural setting. Use of spore-only inoculum typically results in a relatively slow progression of infection compared to inoculum containing active mycelia, which can enhance stochasticity of the infection, particularly at early stages which are characteristically asynchronous. Another major difference was that the previous study evaluated colonization at 9 weeks post infection, a time point at which colonization levels are often in decline. Factors such as the nature/efficacy of the inoculum and the plant growth conditions can greatly influence the infection dynamic and the level of biological variation—concerns that can be addressed through a time course analysis. Our conclusions are based on two mutant alleles in two different mutant backgrounds, each tested at four different timepoints ranging over a one-month period during which colonization was still increasing, reaching near maximal levels (~70% arbuscule occurrence), or had stabilized. A review of transcriptomic data from several different studies failed to support a role for NIN in mycorrhization, as no significant difference was found in *NIN* expression among infected and non-infected cells and between inoculated and non-inoculated cells at different time points. Furthermore, *NIN* has very low transcript levels in non-symbiotic roots and responds to compatible rhizobia with a large and rapid increase in expression, contrasting with its lack of response to mycorrhizal fungi. This non-responsiveness is consistent with an earlier report from *L. japonicus* that monitored *NIN* expression using qPCR over a four-point time course spanning 4 days to 4 weeks after inoculation with *R. irregularis* [44]. In the Guillotin et al. study, which reported a small increase in *NIN* expression in mycorrhizal roots [29], high nitrogen treatment was used to suppress nodulation, a measure which can be expected to reduce but not eliminate nodulation. Finally, Guillotin et al. [29] presented data for the induction of *NIN* by ‘Myc-LCOs’, an equimolar mixture of LCOs produced in bacteria [45] whose biological relevance is still being debated. Indeed, considering the evidence presented here and studies cited herein, the relevance of ‘Myc-LCOs’ to mycorrhiza is questionable. On this point, it is worth noting that the mixture used contains sulfated LCOs that are structurally very similar to *S. meliloti* Nod factors.

In addition to the data presented here, recent phylogenetic studies also contradict a role for NIN in mycorrhization [24,46]. The weight of phylogenetic research suggests a single phylogenetic origin of nodulation with subsequent losses. This was evidenced by the loss of several nodulation-specific genes, including *NIN*, in almost all the non-nodulating plant species tested, despite the fact that the majority of these are able to form symbiosis with AM fungi [24,25]. This suggests that NIN’s role in nodulation is highly specific and refutes a direct role in mycorrhization.

## 4. Materials and Methods

### 4.1. Plant Material

*M. truncatula* Jemalong A17 [47] and *M. truncatula* ssp. *tricycla* R108 seedlings [48] were used in this study. The *nin-1* allele is the result of EMS mutagenesis in the WT A17 background and has an 11bp deletion starting at position 1850. The *nin-2* allele is a *Tnt1* transposon insertion line in the R108 WT background with the insertion lying 20bp upstream of the ATG [10].

### 4.2. Seedling Germination

Seed pods were collected from mature dried *M. truncatula* plants. These were dried for 3–7 days in a 37 °C incubator. The seeds were extracted by crushing the pods with wooden blocks covered in corrugated rubber. The scarification and sterilisation of seeds were performed as described previously [30]. The seeds were then plated on Distilled Water Agar (DWA) plates and inverted to allow for downward root growth. Seeds were put in the dark at 4 °C for stratification for 7 days before transplanting to a soil substrate.

### 4.3. Production of Mycorrhizal Inoculum

To produce an inoculum free from contaminating microbes, particularly rhizobia, all working surfaces, trays, instruments etc. are washed down with 70% ethanol and ethanol sterilized gloves were worn. Terragreen (Oil-Dri Ltd., Wisbech Cambs, UK), sharp sand and Levington’s F1 low nutrient compost (Scotts, Suffolk, UK) at a ratio of 2:2:1 was autoclaved. The seed trays were filled with this soil substrate and approximately 300 chive seeds were evenly distributed on the surface and then lightly covered with soil. This was watered well with dH_2_O and a plastic transparent lid was applied to prevent cross-contamination by rhizobia in the growth rooms. One week after sowing, each germinated plant was inoculated at the base of the stem with 200 sterile *Rhizophagus irregularis* spores DAOM 197198 purchased from Symplanta (Darmstadt, Germany). The plants were grown for eight weeks in the same conditions as the *M. truncatula* plants, being watered to prevent the soil substrate from drying out. After eight weeks, the chive shoots were removed, and the soil substrate/chive root mixture left was then mixed by hand to create a homogenous mixture. To reduce the risk of rhizobial contamination, the inoculum was then transferred into sealed plastic bags. To increase the inoculum in a sterile manner the previously made chive inoculum was evenly spread on the bottom of a seed tray, ensuring an even distribution of chive roots, using 20% of the volume of the tray. A 1:1 mixture of Terra green and sharp sand was added on top (80% of total soil volume) and chive seeds were sown with growth conditions as before. The inoculum was then harvested eight weeks after planting as described above and stored in the dark at 4 °C in sealed plastic bags.

### 4.4. Nodulation

For nodulation, *Sinorhizobium meliloti* Rm1021 strains were grown overnight at 28 °C with shaking. The seedlings were inoculated with 1 mL of rhizobia at a final absorbance of 0.02 at OD_600_ diluted in water one day after transplanting.

### 4.5. Mycorrhization

To compare the differences in arbuscule formation by the fungus *R. irregularis*, the seedlings were germinated as described. The seedlings were then transferred to 1:1 terragreen:sharp sand low nutrient growth medium mixed with 20% chive inoculum containing roots of chive plants infected with spores of *R. irregularis*. The plants were covered with a lid for the first week only to maintain humidity and then were allowed to grow for 2 to 4 weeks before harvesting the root tissue. The roots were washed and approximately one inch of each sample from around two thirds of the total root length was collected for analysis. The fungus was visualized using an ink staining protocol [49]. The arbuscule frequency was then scored using the gridline intersect method [50] under a Nikon Eclipse E800 light microscope with a Pixera Pro 600ES camera.

## Figures and Tables

**Figure 1 plants-09-00071-f001:**
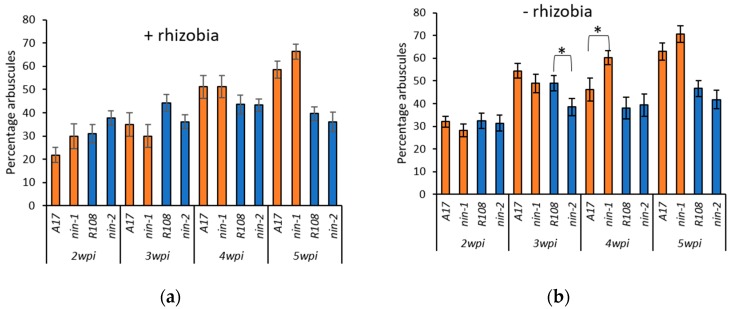
AM phenotype of *nin* mutants with and without addition of rhizobia. (**a**) *M. truncatula* plants were inoculated with *R. irregularis* (10% chive root inoculum) and *S. meliloti* Rm1021 or (**b**) with *R. irregularis* only. The percentage of arbuscules in plants was determined at 2, 3, 4, and 5 weeks post inoculation (wpi). A17 is the WT background for *nin-1*, and *nin-2* is in the R108 background. Bars represent the standard error of the mean. Asterisk indicates a significant difference between the indicated means (Student’s *t*-test, * *p* < 0.05).

**Figure 2 plants-09-00071-f002:**
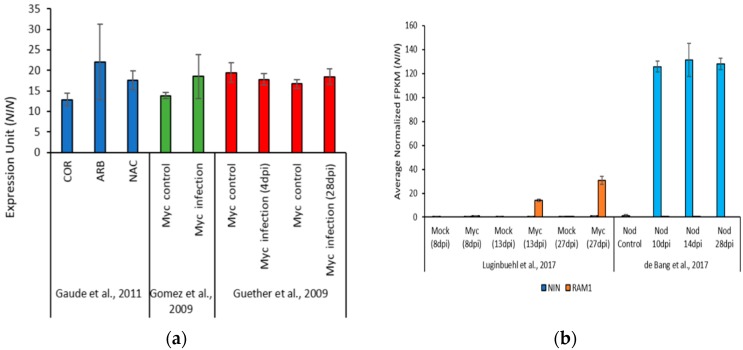
Expression of *Nodule Inception* (*NIN*) during AM symbiosis and nodulation. Quantification of *NIN* expression (**a**) using publicly available transcriptome data (microarray). Raw data were retrieved from the *Medicago truncatula* Gene Expression Atlas [32] and the *Lotus japonicus* Gene Expression Atlas [33]. Data from three independent experiments are shown. The chart on the left shows *NIN* expression in different cell types of *M. truncatula*, collected using laser capture microdissection (LCM) 3 weeks after inoculation with *R. irregularis*. Data are from Gaude et al. [34]. In Gomez et al. [35] chart, *NIN* expression in 6-week-old *M. truncatula* uninoculated and *R. irregularis* inoculated roots is shown. Guether et al. [36] data represent *NIN* expression in uninoculated and *Gigaspora margarita* inoculated roots of *Lotus japonicus* at two different time points 4- and 28-days post inoculation (dpi). COR-cortical cells of non-mycorrhizal roots, ARB-arbuscule-containing cells and NAC-non-colonized cortical cells of mycorrhizal roots. Error bars represent standard error of the mean; (**b**) Quantification of *NIN* expression using publicly available transcriptome data (RNAseq). Raw data were retrieved from *M. truncatula* Small Secreted Peptide Database [39]. The expression of *NIN* and *RAM1* at different time points post inoculation with *R. irregularis* along with their respective mock inoculated controls was compared; data are from Luginbuehl et al. [38]. The expression of *NIN* and *RAM1* at varying developmental stages of nodulation was compared; data are from de Bang et al. [37]. Error bars represent standard error of the mean.

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
