# Peer review of "Nodule Inception Is Not Required for Arbuscular Mycorrhizal Colonization of Medicago truncatula"

_plants, 2020, doi:10.3390/plants9010071_

Round 1

Reviewer 1 Report

The manuscript by Kumar et al is a brief note aiming at the demonstration that Medicago truncatula NIN protein, a key transcription factor for nodule development in Rhizobium-legume symbiosis, is not required for colonization of arbuscular mycorrhiza. The paper is well written, and the point is relevant, given the interest of defining common and different components involved in pathways for the development of both symbiosis. I have some concerns:

1.- Some background information on results similar to those shown in the paper is missing. As cited in the review by Geurts et al (Trends in Plant Science, 2016) the lack of a mycorrhization phenotype of the two nin mutant alleles used was already indicated in other works (including reference 11, not used to illustrate this point.

2.- The Results section presents one experiment in which the authors find that nin mutant allelles are not associated to changes in the colonization efficiency by AM. These data are contradictory with other previously published (ref. 29), and the authors indicate the higher level of mycorrhiza inoculum as a reason to explain the difference with the previous study showing a contradictory result. To say that, the level of inoculum, in spores per plant must be included. Along the same line, the authors should comment on whether a high inoculum level represents a more physiological condition, particularly since previous reports (cited in Delaux et al., 2013) indicate that at high inoculum levels some factors that participate in both the mycorhization and Rhizobium interaction lose the effect on mycorhizal symbiosis.

The authors also present a meta-analysis of NIN expression taking data from several expression databases. In this analysis, the authors conclude that NIN expression is not induced by mycorrhizae infection. The authors should discuss contradictory data in Ref 29 showing that mycorrhized roots, and addition of Myc-LCOs, do induce NIN expression.  

Legend of figure 1: It should be described in a different way assigning specific data to each of the histograms instead of saying “half the plants were also inoculated.

References: not uniform inclusion of doi in all references, with irregular denomination (Doi/doi). Reference 3 has a duplication on the journal’s name.

Author Response

The manuscript by Kumar et al is a brief note aiming at the demonstration that Medicago truncatula NIN protein, a key transcription factor for nodule development in Rhizobium-legume symbiosis, is not required for colonization of arbuscular mycorrhiza. The paper is well written, and the point is relevant, given the interest of defining common and different components involved in pathways for the development of both symbiosis. I have some concerns:

1.- Some background information on results similar to those shown in the paper is missing. As cited in the review by Geurts et al (Trends in Plant Science, 2016) the lack of a mycorrhization phenotype of the two nin mutant alleles used was already indicated in other works (including reference 11, not used to illustrate this point.

We thank reviewer 1 for their time and effort. We have added the appropriate supporting references as suggested, including the review.

2.- The Results section presents one experiment in which the authors find that nin mutant allelles are not associated to changes in the colonization efficiency by AM. These data are contradictory with other previously published (ref. 29), and the authors indicate the higher level of mycorrhiza inoculum as a reason to explain the difference with the previous study showing a contradictory result. To say that, the level of inoculum, in spores per plant must be included. Along the same line, the authors should comment on whether a high inoculum level represents a more physiological condition, particularly since previous reports (cited in Delaux et al., 2013) indicate that at high inoculum levels some factors that participate in both the mycorhization and Rhizobium interaction lose the effect on mycorhizal symbiosis.

We have changed the wording, we no longer call our inoculum ‘stronger’, but instead describe it in detail:

“The differences could be accounted for by the different types of inoculum used. The previous study used a fixed quantity of spores as inoculum, while the inoculum used for this study was a fresh inoculum containing spores, mycelia and colonized root fragments which presumably more closely resembles what plants encounter in their natural setting. Use of spore-only inoculum typically results in a relatively slow progression of infection compared to inoculum containing active mycelia, which can enhance stochasticity of the infection, particularly at early stages which are characteristically asynchronous.”

We have left judgement of what is ‘strong’ and ‘weak’ to the interpretation of the reader, but presumably under natural conditions plants will generally encounter a mixture of spores and mycelia rather than spores alone, the former presumably being more efficient.

The authors also present a meta-analysis of NIN expression taking data from several expression databases. In this analysis, the authors conclude that NIN expression is not induced by mycorrhizae infection. The authors should discuss contradictory data in Ref 29 showing that mycorrhized roots, and addition of Myc-LCOs, do induce NIN expression. 

We now address this issue directly in the discussion, expressing our opinion that using high N is not an effective safeguard against rhizobia contamination/ nodulation. High N does of course suppress nodulation quite well, but in such experiments contamination by rhizobia is common, resulting a few small bumps, which may explain why they saw this slight increase in NIN expression. Indeed, Guillotin et al mentioned one experiment in the GeneAtlas that showed a similar increase in NIN in AM roots, but learned that this was in fact due to rhizobial contamination (Pascal Ratet, personal communication). This and the issue of Myc-LCOs are now discussed as follows:

"Furthermore, NIN has very low transcript levels in non-symbiotic roots and responds to compatible rhizobia with a large and rapid increase of expression, contrasting with its lack of response to mycorrhizal fungi. This non-responsiveness is consistent with an earlier report from L. japonicus that monitored NIN expression using qPCR over a 4-point time course spanning 4 days to 4 weeks after inoculation with R. irregularis [43]. In the Guillotin et al. study, which reported a small increase in NIN expression in mycorrhizal roots [29], high nitrogen treatment was used to suppress nodulation, a measure which can be expected to reduce but not eliminate nodulation. Finally, Guillotin et al. [29] presented data for induction of NIN by ‘Myc-LCOs’, an equimolar mixture of LCOs produced in bacteria [44] whose biological relevance is still being debated. Indeed, considering the evidence presented here and studies cited herein, the relevance of ‘Myc-LCOs’ to mycorrhiza is questionable. On this point, it is worth noting that the mixture used contains sulfated LCOs that are structurally very similar to S. meliloti Nod factors"

Legend of figure 1: It should be described in a different way assigning specific data to each of the histograms instead of saying “half the plants were also inoculated.

References: not uniform inclusion of doi in all references, with irregular denomination (Doi/doi). Reference 3 has a duplication on the journal’s name.

These issues have been addressed.

Reviewer 2 Report

A solid piece of work. Well done

I feel the paper is utterly well written and does need any further
improvement. Also the experiments are well described and conducted
very well. So I do not think that the authors need to put extra energy
into this paper.

Author Response

Thanks for supporting our manuscript

Reviewer 3 Report

Dear Authors,

thank you for the opportunity to review your paper on NIN role in arbuscular mycorrhiza in Medicago truncatula. Generally, I find the results interesting. The manuscript contains some minor errors, and in some places additional details are needed - both the errors/inconsistencies as well as details lacking are marked in the annotated manuscript.

My major concern is your discussion of the Guillotin's team paper - I find this discussion insufficient and a bit biased, and therefore I recommend major correction. My comment concerning this issue you will find in the annotated manuscript.

With best regards,

yours sincerely,

reviewer

Author Response

Dear Authors,

thank you for the opportunity to review your paper on NIN role in arbuscular mycorrhiza in Medicago truncatula. Generally, I find the results interesting. The manuscript contains some minor errors, and in some places additional details are needed - both the errors/inconsistencies as well as details lacking are marked in the annotated manuscript.

My major concern is your discussion of the Guillotin's team paper - I find this discussion insufficient and a bit biased, and therefore I recommend major correction. My comment concerning this issue you will find in the annotated manuscript.

With best regards,

yours sincerely,

We thank reviewer 3 for their detailed/annotated comments, all of which we have addressed in the updated manuscript. Most importantly, we have provided a detailed review of the Guillotin data in context of our findings (and the additional reports brought to our attention by reviewer 1). The discussion has been updated as follows:

“While our conclusion regarding the absence of an AM phenotype is consistent with earlier report on nin in L. japonicus [20,42], it contradicts an earlier study that found that M. truncataula nin-1 had strongly reduced AM colonization at nine weeks post inoculation, and had a mild decrease in penetration events at two weeks post inoculation [29]. The differences could be accounted for by the use of different types of inoculum used. The previous study used a fixed quantity of spores as inoculum, while the inoculum used for this study was a fresh inoculum containing spores, mycelia and colonized root fragments which presumably more closely resembles what plants encounter in their natural setting. Use of spore-only inoculum typically results in a relatively slow progression of infection compared to inoculum containing active mycelia, which can enhance stochasticity of the infection, particularly at early stages which are characteristically asynchronous.  Another major difference was that the previous study evaluated colonization at 9 weeks post infection, a time point at which colonization levels are often in decline. Factors such as the nature/efficacy of the inoculum and the plant growth conditions can greatly influence the infection dynamic and the level of biological variation—concerns that can be addressed through a time course analysis. ”

Reviewer 4 Report

The submitted manuscript by Kumar et al investigates the mycorrhizal colonization phenotype in different nin mutant genetic backgrounds by analyzing AM arbuscule formation at different time points.

The role played by NIN during SNF has been studied in a plethora of reports and recently further developed by a transcriptomic analysis aimed to identify the NIN molecular targets. The submitted article provides a little but significant progress about the investigation of the NIN functions, clearly stating that it is not involved in Mycorrhizal colonization.

Author Response

We thank the reviewer for their support.

Round 2

Reviewer 3 Report

Dear Authors,

I greatly appreciate your effort put into improving the manuscript, especially the Discussion, and I believe that the text has gained greater clarity and depth of thought.

However, you omitted addressing an issue, which I considered important, and I marked it in the 1st version of your manuscript. Therefore, I return to the problem in this round of review:

The WT plants should not only be of the “wild” genotype, but also they should be grown under optimal conditions to provide the correct background for comparisons, and in case of M. truncatula, plant’s fitness depends on the efficiency of two symbioses. You devoted much effort to create favorable conditions for the development of mycorrhiza – I mean the preparation of mycorrhizal inoculum – then why no such effort was undertaken to provide favorable conditions for the efficient N2 fixation? The efficiency of host's diazotrophic symbiosis with rhizobia affects host's general fitness, and indirectly affects also host's ability to feed photosynthates to mycorrhizal symbiont. It really has to be explained (in M&M chapter would be OK), why a rhizobial strain, which was proven to be ineffective N2 fixer both on A17 as well as on R108 M. truncatula genotypes (the Terpollilli’s and Kazmierczak’s teams papers I indicated previously) was used to produce WT plants for comparisons with nin mutants.

Additionally, some M&M information concerning growth conditions should be added, as well as some editorial issues should be corrected within the text and references, and they are marked in the manuscript annotated pdf file.

With best regards, and best wishes for the new year :-)

rev

Author Response

We thank reviewer three for their additional efforts towards improving our manuscript. We have made the suggested changes in the revised manuscript.

Regarding the choice of Rm1021, we agree completely that it is a sub-optimal symbiont for Medicago truncatula, and in the future will consider this when choosing strains. Despite this, this strain does effectively nodulate Medicago and has been used extensively for research on nodulation. In this case we did check to be sure nodulation occurred, so we expect if nodulation had a large impact on mycorrhizal colonization of nin we would have been able to detect it.